# Robot-Assisted Magnetic Resonance Imaging-Targeted versus Systematic Prostate Biopsy; Systematic Review and Meta-Analysis

**DOI:** 10.3390/cancers15041181

**Published:** 2023-02-13

**Authors:** Vladislav Petov, Camilla Azilgareeva, Anastasia Shpikina, Andrey Morozov, German Krupinov, Vasiliy Kozlov, Nirmish Singla, Juan Gómez Rivas, Moreno-Sierra Jesús, Stefano Puliatti, Enrico Checcucci, Severin Rodler, Ines Rivero Belenchon, Karl-Friedrich Kowalewski, Alessandro Veccia, Jonathan Mcfarland, Giovanni E. Cacciamani, Mark Taratkin, Dmitry Enikeev

**Affiliations:** 1Institute for Urology and Reproductive Health, Sechenov University, 119991 Moscow, Russia; 2Department of Public Health and Healthcare, Sechenov University, 119991 Moscow, Russia; 3Departments of Urology and Oncology, The James Buchanan Brady Urological Institute, Johns Hopkins University School of Medicine, Baltimore, MD 21287, USA; 4Department of Urology, Clinico San Carlos University Hospital, 28040 Madrid, Spain; 5Young Academic Urologists (YAU) Working Group in Urotechnology of the European Association of Urology, 30016 Arnhem, The Netherlands; 6Urology Department, University of Modena and Reggio Emilia, 42121 Modena, Italy; 7Department of Surgery, Candiolo Cancer Institute, FPO-IRCCS, Candiolo, 10060 Turin, Italy; 8Department of Urology, Klinikum der Universität München, 81377 Munich, Germany; 9Department of Uro-Nephrology, Virgen del Rocío University Hospital, 41013 Seville, Spain; 10Department of Urology, University Medical Center Mannheim, Heidelberg University, 69047 Heidelberg, Germany; 11Urology Unit, Azienda Ospedaliera Universitaria Integrata, 37126 Verona, Italy; 12Institute of Linguistics and Intercultural Communication, Sechenov University, 123242 Moscow, Russia; 13Faculty of Medicine, Universidad Autónoma Madrid, 28029 Madrid, Spain; 14Catherine and Joseph Aresty Department of Urology, Keck School of Medicine, University of Southern California, Los Angeles, CA 90033, USA; 15Artificial Intelligence Center at USC Urology, USC Institute of Urology, University of Southern California, Los Angeles, CA 90033, USA; 16Department of Urology, Medical University of Vienna, 1090 Vienna, Austria

**Keywords:** systematic review, meta-analysis, robot-assisted prostate biopsy, MR-targeted biopsy, prostate cancer

## Abstract

**Simple Summary:**

Robotic devices are being actively introduced into the urology field. We aimed to assess the diagnostic performance and safety of robot-assisted targeted (RA-TB) and systematic prostate biopsies (RA-SB). RA-TB and RA-SB are both technically feasible and have comparable csPCa and overall detection rates [OR = 1.02 (95% CI 0.83; 1.26), *p* = 0.05, I^2^ = 62% and OR = 0.95 (95% CI 0.78; 1.17), *p* = 0.17, I^2^ = 40%, respectively]. A robot-assisted biopsy can potentially be performed under local anesthesia or sedation. Furthermore, a robot-assisted biopsy is a safe and feasible option with a low risk of complications.

**Abstract:**

Introduction: Robot-assisted devices have been recently developed for use in prostate biopsy. However, it is possible advantages over standard biopsy remain unclear. We aimed to assess the diagnostic performance and safety of robot-assisted targeted (RA-TB) and systematic prostate biopsies (RA-SB). Methods: A systematic literature search was performed in MEDLINE and Scopus databases. The detailed search strategy is available at Prospero (CRD42021269290). The primary outcome was the clinically significant prostate cancer (PCa) detection rate. The secondary outcomes included the overall detection rate of PCa, cancer detection rate per core, and complications. Results: The clinically significant cancer detection rate, overall cancer detection rate, and “per patient” did not significantly differ between RA-TB and RA-SB [OR = 1.02 (95% CI 0.83; 1.26), *p* = 0.05, I^2^ = 62% and OR = 0.95 (95% CI 0.78; 1.17), *p* = 0.17, I^2^ = 40%, respectively]. There were no differences in the clinically insignificant cancer detection rate “per patient” between RA-TB and RA-SB [OR = 0.81 (95% CI 0.54; 1.21), *p* = 0.31, I^2^ = 0%]. RA-TB had a significantly higher cancer detection rate “per core” [OR = 3.01 (95% CI 2.77; 3.27), *p* < 0.0001, I^2^ = 96%]. Conclusion: RA-TB and RA-SB are both technically feasible and have comparable clinical significance and overall PCa detection rates.

## 1. Introduction

Prostate biopsy remains the main method for the definitive diagnosis of prostate cancer (PCa) despite the active development of new biomarkers and imaging modalities [1]. More than one million prostate biopsies are obtained annually [2]. A ultrasound guided transrectal systematic prostate biopsy has been the standard method for diagnosing PCa for nearly 30 years, but the number performed has been steadily declining [3,4]. This is due to European Association of Urology guidelines which suggest that we should be able to rely on the transperineal approach as it results in a lower rate of postoperative infectious complications [5,6]. However, independently of transrectal or transperineal approaches, US biopsies are characterized by overdiagnosis of clinical insignificant (cis) PCa [7]. In this regard, there is an increased interest in MR-targeted biopsy to eliminate this drawback, with the so-called magnetic resonance imaging (MRI) pathway as a means of PCa diagnosis [4,8]. This implies using multiparametric MRI prior to the first biopsy, and if it is positive (i.e., PI-RADS ≥ 3), then a MR-targeted biopsy must be completed [1].

Among the major types of MR-targeted biopsy (TB), we find the in-bore MR-TB (MRI-TB), MRI-TRUS fusion TB (FUS-TB), and cognitive TB (COG-TB) methods. Each of these demonstrates a high but clinically comparable, significant (cs), and overall cancer detection rate (CDR) [9,10]. Being an effective and safe procedure, transperineal biopsy still has a number of limitations; among them is the dependence on the surgeons’ learning curve and the increased (compared to the transrectal route) operation time [11,12,13,14]. This has led to the development of robot-assisted biopsy devices [15].

Robot-assisted biopsy devices utilize a robotic arm that provides precise three-dimensional (3D) targeting of the biopsy needle that determines the needle direction, penetration depth, and biopsy position and allows all biopsies to be taken from the same incisions and defines the penetration depth automatically [16]. The preliminary results of the trials showed that they were sufficiently safe and able to overcome the previously mentioned limitations. However, the exact role of robot-assisted systems in biopsy is still far from clear. This systematic review and meta-analysis aim to summarize the available data on robotic-assisted MR-targeted prostate biopsy and compare the overall and clinically significant cancer detection rate of robot-assisted targeted (RA-TB) and systematic prostate biopsies (RA-SB).

## 2. Materials & Methods

### 2.1. Search Strategy and Inclusion Criteria

This systematic review was carried out in accordance with the Preferred Reporting Items for Systematic Reviews and Meta-Analyses (PRISMA) guidelines (see PRISMA statement, Figure 1). The detailed search strategy and review protocol has been published in Prospero (CRD42021269290).

The scope of the review according to the PICOS process (Patient, Intervention, Comparison, Outcomes, Studies) is as follows:P—Patients with suspected PCa or on active surveillance;I—Robot-assisted MRI-targeted prostate biopsy;C—Systematic or MRI-targeted biopsy (with and without robot assistance);O—Detection rate of csPCa, overall detection rate of PCa, cancer detection rate per core, and complication rate;S—Both prospective and retrospective studies.

A systematic literature search was performed using two databases (Medline (PubMed) and Scopus) and the following search parameters: robot* AND biopsy AND prostate NOT prostatectomy. We did not include, on purpose, in the search other terms such as «MRI-guided», «fusion», «targeted», and so on because we wanted to be consistent in our use of terminology. We believed this broad search would allow us not to miss any data related to robot-assisted biopsy. As this procedure is not widely used, it would not be difficult to remove manually any papers that might be considered irrelevant.

The inclusion criteria were as follows: all types of studies (both prospective and retrospective) report their own data on robotic biopsy accuracy and complication rates. We included only articles in English from the last 10 years. Any other literature without original data or sufficient information were excluded. These might include different types of reviews, comments, single cases, editorial material, and books as well as conference abstracts.

Firstly, VP performed a title review. He excluded any publications that did not fit the aforementioned criteria. Secondly, VP and AS independently performed abstract reviews according to the same criteria. In the event of disagreement between the reviewers at this stage, the relevant article would be included for further full-text review. Once the title and abstract were reviewed, AM manually removed any duplicates. As a last step, VP, AS, and AM independently performed a full-text review. In the event of any disagreement, each party made their case and tried to resolve it. If they could not come to an agreement, DE made the final decision.

### 2.2. Studies’ Quality Assessment

The risk of bias was assessed using QUADAS−2 tool https://www.bristol.ac.uk/population-health-sciences/projects/quadas/quadas-2/; accessed on 10 July 2021.

### 2.3. Data Extraction

The following raw data were extracted manually from the articles: number of patients, age, PSA level, prostate volume, flux density (tesla), Pi-RADS score, number of lesions, robot-assisted targeted biopsy (RA-TB) type, software, hardware, approach, number of cores, comparative biopsy (if applicable), the cancer detection rate of csPCa, the overall cancer detection rate (CDR), CDR per core, and CDR per lesion. The data include both the biopsy naïve and those who had a prior negative biopsy. The data regarding patients on active surveillance were excluded.

The primary outcome was the detection rate of csPCa. The secondary outcomes included the overall detection rate of PCa, the cancer detection rate per core, and complications.

### 2.4. Methods of Meta-Analysis

The statistical heterogeneity of the results included in the meta-analysis was evaluated using χ^2^ and the heterogeneity index I^2^. To assess OR, Mantel–Haenszel test was used. The heterogeneity threshold was defined as 50% using Q test. Where there was significant heterogeneity, a random-effects model was applied, while a fixed effects model was used in the absence of heterogeneity. We tested the generalized point estimation of the effect and its confidence limits. To visualize the results of the analysis, we used forest and funnel plots. RevMan 5.3 software was used for data processing.

## 3. Results

During the full-text review, 10 articles were excluded. The work of Ho et al. was excluded as only robotic-assisted systematic prostate biopsy was performed [17]. In a study by Perlis et al., the positioning of the ultrasound sensor was performed using the fusion Bx semi-robotic arm, which does not fully meet the criteria for a robotic system [18]. There were eight studies that aimed to assess the utility and safety of RA-MRI-TB [19,20,21,22,23,24,25,26], two of which also compared it to MRI-TB with manual assistance [20,26]. These studies were not included in the final analysis due to the absence of a comparator in the form of RA-SB (PRISMA statement, Figure 1). Finally, we selected nine studies [16,27,28,29,30,31,32,33,34], five of which were prospective (none of them was randomized), and four were retrospective. Four studies with a total of 742 patients were included in the meta-analysis [16,28,30,32]. Seven studies assessed transperineal RA-FUS-TB compared to RA-SB [16,28,30,31,32,33,34], one study assessed RA-COG-TB compared to RA-SB [27], and one study assessed transperineal RA-FUS-TB robot-assisted fusion biopsy compared to COG-TB and MRI-TB [29].

Six studies had a low risk of bias, two showed intermediate risk of bias (identified as “some concerns”), and one showed a high risk of bias according to QUADAS−2 (Figure 2).

### 3.1. Demographics

The mean pooled age was 66 ± 7.7. The mean pooled PSA level was 9.6 ± 7.3 ng/mL. The mean pooled prostate volume was 42.1 ± 20.40. The number of biopsy-naïve patients was 418, and the repeated biopsy was performed in 324 patients. Three papers fully reported PI-RADS scores [16,30,32]. PI-RADS 1–2—67 lesions, PI-RADS 3—165, PI-RADS 4—401 lesions, and PI-RADS 5—137 lesions. Median or mean number of cores for RA-TB ranged from 1.5 to 13.3 and from 14.0 to 32.0 for RA-SB.

CsPCa was variously defined by the authors: (1) any Gleason score > 6; (2) any percentage of a Gleason 4 in a biopsy core; (3) Gleason score > 6 or maximum cancer core length (MCCL) > 3 mm for Gleason 6; (4) Gleason score 7 or MCCL ≥ 5 mm for Gleason 6; and (5) Gleason score 6 with > 1 positive core or MCCL > 6 mm (defined in detail for each trial in Table 1).

All RA-FUS-TBs were performed via transperineal approach using the iSR’obot Mona Lisa™ (Biobot Surgical, Singapore).

### 3.2. Cancer Detection Rate (CDR) “per Patient”

The csPCa and overall CDR “per patient” did not significantly differ between RA-TB and RA-SB [OR = 1.02 (95% CI 0.83; 1.26), *p* = 0.05, I^2^ = 62% and OR = 0.95 (95% CI 0.78; 1.17), *p* = 0.17, I^2^ = 40%, respectively] (Figure 3, Figure 4, Figure 5 and Figure 6). There were no differences in the cisCDR “per patient” between RA-TB and RA-SB [OR = 0.81 (95% CI 0.54; 1.21), *p* = 0.31, I^2^ = 0%] (Figure 7 and Figure 8).

Most studies revealed that there was no significant difference between RA-TB and comparative biopsy in terms of the csPCa detection rate. As for the detection rate, seven out of seventeen studies include comparative data on the overall detection rate of PCa.

### 3.3. Additional Utility

With regards to csPCa, the additional utility of RA-SB varied from 1.8% and 3.9% to 7.5% and 10.4% [16,28,32,33]. Kauffmann et al. and Mischringer et al. reported similar results on the additional utility of RA-SB with regards to the overall CDR: 9.1 and 13.8%, respectively [28,32]. Lee et al. reported results on the additional utility of RA-SB with regards to the overall CDR of 8.1% and csPCa of 7.6% [30].

### 3.4. Cancer Detection Rate “per Core”

RA-TB had a significantly higher cancer detection rate “per core” [OR = 2.08 (95% CI 1.30; 3.30), *p* = 0.002, I^2^ = 95%] (Figure 9 and Figure 10). Only three studies on RA-TB and one study on in-bore MRI-guided biopsy analyzed the cancer detection rate per core [20,27,28,32].

In the study of Mischringer et al., the CDR per core of RA-TB was 27%, whereas that of RA-SB was only 11% [32]. In another study, the csPCa per core detected by RA-TB and RA-SB was 41 and 5%, respectively [28]. Lee et al. reported that the CDR per core of RA-TB was 23.2% while that for RA-SB was 9.8% [30].

### 3.5. Cancer Detection Rate “per Prior Negative Patient”

The cancer detection rate “per prior negative patient” did not significantly differ between RA-TB and RA-SB [OR = 1.09 (95% CI 0.76; 1.56), *p* = 0.65, I^2^ = 55%] (Appendix A). The csCDR “per prior negative patient” also did not significantly differ between groups [OR = 1.18 (95% CI 0.81; 1.72), *p* = 0.39, I^2^ = 72%] (Appendix A). There were also no differences in the cisCDR per prior negative patient [OR = 1.00 (95% CI 0.64; 1.55), *p* = 0.46, I^2^ = 0%] (Appendix A).

### 3.6. Anesthesia

General anesthesia was used in 7 out of 17 studies. Yang et al. were the only authors who reported the use of local anesthesia and sedation when performing combined transperineal RA-TB and RA-SB [34].

### 3.7. Procedure Time

The duration of RA combined (RA-FUS-TB + RA-SB) biopsy ranged from 19 to 49 min [24]. However, the time decreased with each subsequent patient. (The median duration of RA-SB was 33 min (range 19–49 min) [24]).

### 3.8. Complications

Only two studies reported complications according to the Clavien–Dindo classification [32,34]. In the Mischinger et al. study, the rate of minor (I–II) and major (≥III) complications was 3 and 1.5%, respectively [32]. Yang et al. found 6.7% of Clavien–Dindo II complications [34]. In the other studies, most of the complications included re-catheterization (5.4–12.8%), hematuria (2.3–9.1%), and rectal bleeding (1.9–3.3%).

## 4. Discussion

The meta-analysis confirmed that RA-TB and RA-SB resulted in a similar detection rate of PCa. Both techniques were comparable in terms of csPCa as well as for the overall PCa detection rate. However, the number of cores obtained by TB was 2–3 times less than SB. These results are in line with those of the MRI-FIRST and 4 M studies where there was no statistically significant difference between targeted and systematic biopsies in terms of the csPCa detection rate [35,36]. Different results were achieved in the PRECISION trial, which showed that the MRI-pathway was superior to the standard approach. We see two possible explanations for this difference: firstly, in the included studies, MRI-targeted and systematic biopsies were performed in the same patients at the same time. This increases the probability of cross-influencing tests, which would make it difficult to evaluate the unbiased performance of each test individually. Secondly, the number of SB cores varied from 14 to 32. In the included studies, the systematic biopsy detection rate was seen to increase, but in the PRECISION trial, it did not exceed 12. However, this needs to be further evaluated. Therefore, we believe that for those who are undergoing robot-assisted biopsy, targeted biopsy should be accompanied with systematic biopsy (in line with the findings of the systematic review by Drost et al.). Such an approach also should be specifically discussed in patients with previous negative biopsies [37]. Despite the fact that, according to the literature, targeted biopsies detect less cisPCa, in our analysis, we did not find any difference [37]. This may be due to the fact that in three studies out of four, the number of targeted cores exceeded four (Table 2).

Currently, there is lack of prospective well-designed studies comparing head-to-head robotic versus non-robotic biopsy, but there is an ongoing trial assessing the TRUS robot and conventional TRUS biopsy [38]. Nevertheless, it is worth discussing robotic biopsy because it has a great potential to shift current standard modalities. Initially, robotic biopsy systems were created to improve needle placement accuracy, standardizing the procedure with reproducible quality, making it independent of skill sets, and reducing procedure time in contrast to FUS-TB, which results depend on the surgeon’s experience [39]. In a preclinical study, Ho et al. presented the concept of the robotic ultrasound-guided prostate biopsy system and evaluated that its repeatable accuracy is lower than 1 mm [15]. Later, Lim et al. proved its accuracy of 1.43 mm [40]. In another preclinical trial on a canine model, Muntener et al. evaluated that the median error for MR-guided RA biopsy was 2.02 mm [41]. The robotic system allows for the reduction of prostate displacement and deformation during the biopsy, which is known to occur during TRUS. It was preclinically evaluated that, during robotic prostate biopsy, the difference between the pre-acquired contour and real-time contour was 0.89 mm [40]. In addition, the robotic system can solve the traveling salesman problem (to find the shortest route that starts from the original position of the probe, visits each biopsy point once, and returns to the original position of the probe). Thus, an in-built robotic system algorithm minimizes probe movement, leading to a reduction in biopsy time (the average time was 4.42 min) [36].

In most studies, both RA-TB and RA-SB were performed [16,27,28,29,30,31,32,33,34,39], which could be crucial for biopsy-naïve patients, as the percentage of csPCa that can be missed by only targeted biopsy ranges from 2.6 to 8.7% [35,36]. When considering focal therapy or active surveillance, the systematic cores may be also added because of the need to rule out multiparametric MRI invisible cisPCa lesions [42]. Ho et al. conclude that robotic prostate biopsy fulfils the framework of focal therapy and might be the platform for further ablative treatment modalities for PCa [17,43]. In addition, this system, as well as those presented by Stoianovici et al., are the only systems with robotic control of the depth of the biopsy needle punctures, ensuring the suspicious lesion is in the middle of the biopsy core, which could be crucial when considering further ablation procedures [33,44].

Moreover, RA prostate biopsies can be performed with only two skin transperineal punctures irrespective of the number of cores taken, which could potentially be performed under local anesthesia alone or with sedation [34]. The visual analogue pain score was 0 in all cases, supporting the fact that the two-puncture approach could be easily tolerated by patients [34]. Wetterauer et al. also used the two-puncture approach, while in their study, the VAS score ranged between 0 and 8 [45]. Local anesthesia could be beneficial for elderly patients or those with comorbidities [34]. In addition, no anesthesiologic assistance is required. Therefore, the procedure could be performed in outpatient settings. However, there were no clinical trials performing RA-FUS-TB under local anesthesia alone.

It is worth mentioning that transperineal RA-TB could be performed without antibiotic usage. Kroenig et al. used no antibiotics prophylaxis in their trial with no infectious complications reported [16]. Wetterauer et al., in 177 patients who underwent transperineal fusion biopsy using a two-puncture approach, used no periprocedural antibiotic prophylaxis with no infectious complications [45]. As there is growing evidence supporting the fact that transrectal biopsy has a higher infectious complication rate, the transperineal two-puncture approach could minimize antibiotic usage and prevent antibiotic resistance [46,47,48].

In-bore biopsy could be an alternative to the fusion approach, but in-bore biopsies were time-consuming procedures, so RA MRI-TB was introduced to reduce operation time. In the initial preclinical trial, the procedure time was estimated at 30 min [49]. However, in early clinical trials, the mean procedure time for the transrectal approach was 76.5 and 76 min [24,26]. However, in later studies, the manipulation time was significantly reduced to 33.9–37.8 min [19,21,23,25,50]. As a final result, the procedure time of a transrectal RA MRI-TB compares with that of a FUS-TB [51]. Nevertheless, one should keep in mind the main disadvantages of the transrectal approach. Where the transperineal approach is concerned, the whole procedure time was initially 141.67 for the RA MRI-TB group [20]. Now, there is a lack of data evaluating the total procedure time for the transperineal approach.

According to our systematic review and meta-analysis, the most common complications among all studies were acute urinary retention, hematuria, hematoma, and rectal bleeding, which were minor and self-limiting. Only two studies reported complications according to the Clavien–Dindo grading system after transperineal RA biopsy, resulting in the fact that almost all complications were Claven–Dindo II (6.7%) or lower, while only one patient had rectal injury and peritonitis (Clavien–Dindo III) [32,34]. These results are in line with those from Wegelin et al.’s FUTURE trial, where the grade two complication rate was 5.1% with no evidence of a higher grade [52]. It should be noted that the transperineal approach is associated with an increased risk of perineal swelling and hematoma compared to the transrectal approach (10.20% vs. 2.08%, *p* = 0.006) [53], whilst the two-needle approach may potentially decrease the risk of perineal hematoma and swelling [32].

There were some limitations in the present study. The principal limitation is the quality and heterogeneity of the studies we included. All of them were non-randomized and mostly with a low number of patients. Secondly, in 2013, Moore et al. established the START criteria (Standards of Reporting for MRI-targeted Biopsy Studies) which aimed to standardize the results [54]. Unfortunately, not all authors strictly adhered to reporting the biopsy results according to the START criteria. We believe that this should be addressed in future trials to enhance the quality of results. Thirdly, systematic biopsies included a various number of cores taken (from 12 to 29), which could affect the overall cancer detection rate and csCDR. The fourth limitation was the inclusion of the patients with both PI-RADS 1–2 and 3–5 lesions. Nevertheless, the lesions with PI-RADS 1–2 were no more than 10%, so we believe that this is unlikely to affect the results. Unfortunately, not all the studies fully reported the PI-RADS scores; therefore, it was not possible to perform a subgroup analysis of CDR using PI-RADS. The fifth limitation was the different definitions of the term “clinically significant prostate cancer” used by the authors. The articles were published in different years, and the definition of csPCa has varied to such an extent that currently there is still no consensus. Last is the absence of uniformity in the results, reporting only one single trial with data on negative predictive values, area under curve, specificity, and sensitivity.

## 5. Conclusions

RA-TB and RA-SB are both technically feasible and have comparable csPCa and overall detection rates. A robotic biopsy system can potentially be performed under local anesthesia or sedation. Robot-assisted biopsy is a safe and feasible option with a low risk of complications.

## Figures and Tables

**Figure 1 cancers-15-01181-f001:**
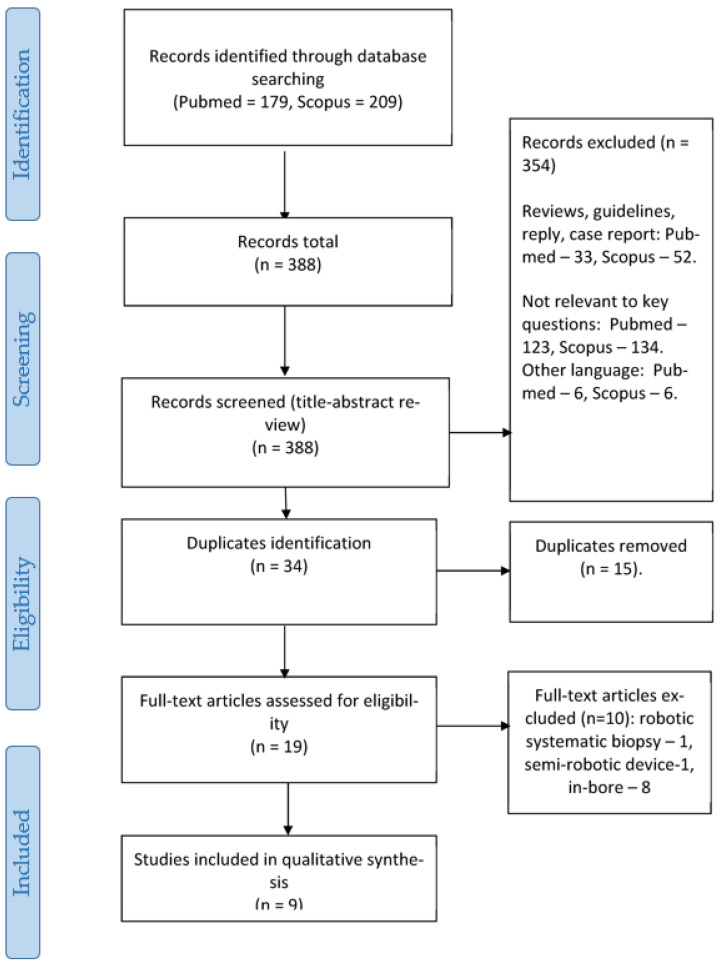
PRISMA flow chart.

**Figure 2 cancers-15-01181-f002:**
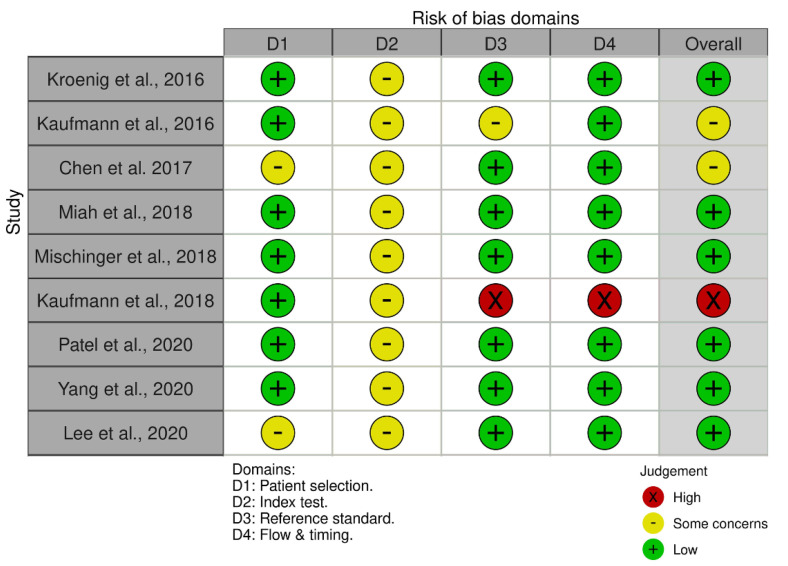
Risk of bias assessed using QUADAS−2. The risk of bias was assessed using QUADAS−2 tool https://www.bristol.ac.uk/population-health-sciences/projects/quadas/quadas-2/, accessed on 10 July 2021.

**Figure 3 cancers-15-01181-f003:**
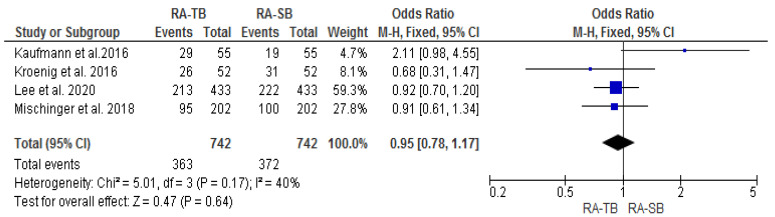
Clinically significant cancer detection rate “per patient”.

**Figure 4 cancers-15-01181-f004:**
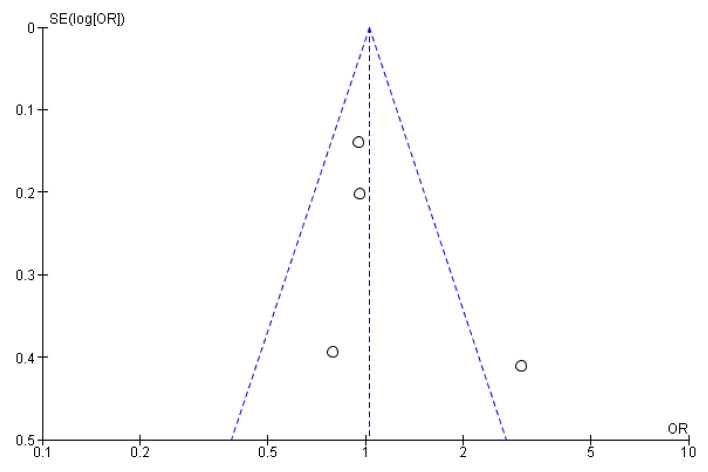
Funnel plot for clinically significant cancer detection rate “per patient”.

**Figure 5 cancers-15-01181-f005:**
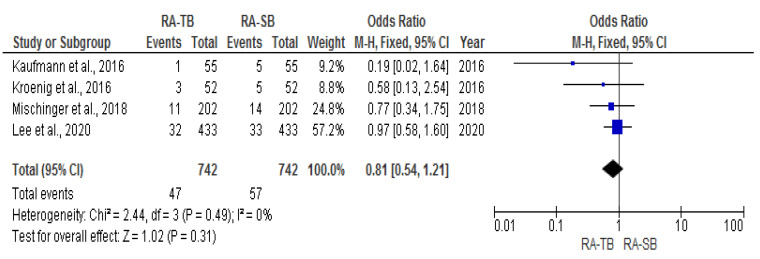
Cancer detection rate “per patient”.

**Figure 6 cancers-15-01181-f006:**
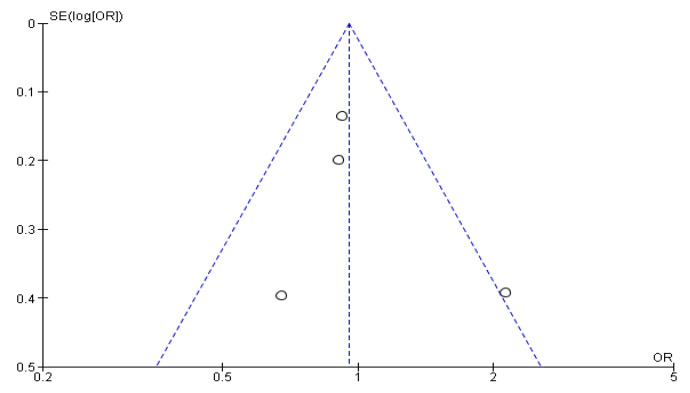
Funnel plot for cancer detection rate “per patient”.

**Figure 7 cancers-15-01181-f007:**
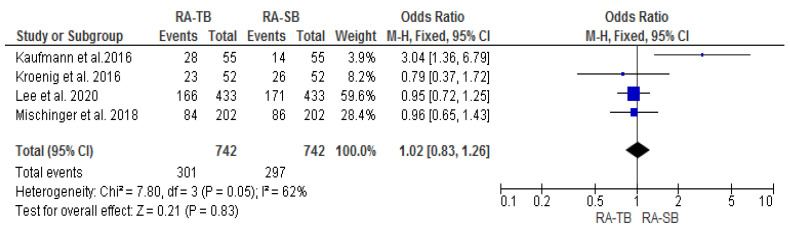
Clinically insignificant cancer detection rate per patient.

**Figure 8 cancers-15-01181-f008:**
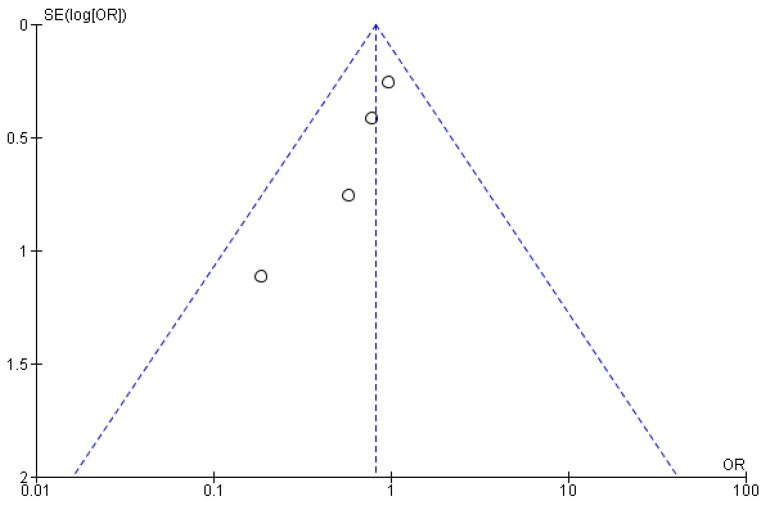
Funnel plot for clinically insignificant cancer detection rate per patient.

**Figure 9 cancers-15-01181-f009:**
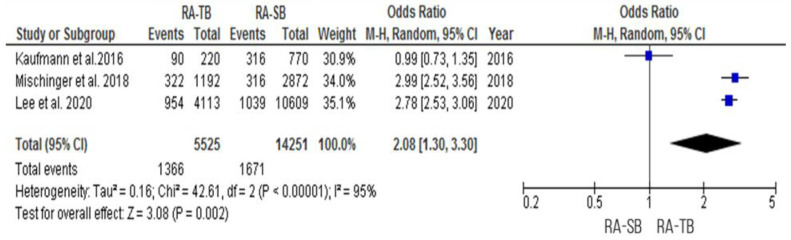
Cancer detection rate “per core”.

**Figure 10 cancers-15-01181-f010:**
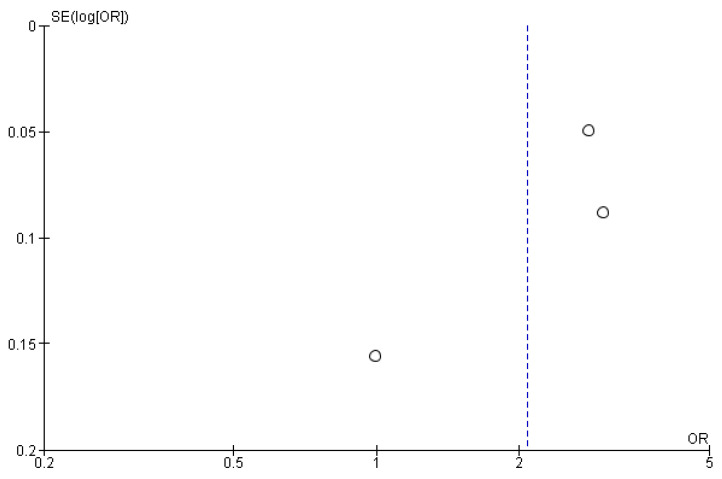
Funnel plot for cancer detection rate “per core”.

**Table 1 cancers-15-01181-t001:** Design, study population, and magnetic resonance imaging.

Author, Year	Type of Study, Level of Evidence	Number, Type of Patients	Age, Years	PSA, ng/mL	Prostate Volume, cm^3^	MRI Flux Density, T	MRI Lesions Targeted	Number of MRI Lesions	Definition of csPCa
Kroenig et al., 2016 [16]	Retrospective, 2b	52—prior negative	Mean 65.8 (±7.3) Median 66.0 (IQR 60.0–71.8)	Mean 9.9 (± 5.9) Median 8.8 (IQR 5.7–13.04)	Mean 57.6 (± 26.6) Median 49.3 (IQR 37.8–73.3)	NA	2–5	PI-RADS unclassified—12 PI-RADS 2—16 PI-RADS 3—55 PI-RADS 4—40 PI-RADS 5—12	Gleason grade ≥ 4
Kaufmann et al., 2016 [28]	Prospective, nonrandomized, 2b	55—prior negative	Mean 65.0 (± 7.9; range 50–78)	Mean10.2 (±5.2; range 3.2–25.1)	Mean 43.9 (± 22.6; range 13.4–108)	1.5 3.0	NA	≥4—24 ≥3—33	GS ≥ 3 + 4
Chen et al., 2017 [27]	Prospective, nonrandomized, 2b	18—active surveillance	Mean 65.4 (±4.9)	Mean 7.0 (± 1.8)	Mean 32.1 (± 13.4)	3.0	≥3	PI-RADS 3—5 PI-RADS 4—7 PI-RADS 5—6	Gleason grade ≥ 4
Miah et al., 2018 [31]	Prospective, nonrandomized, 2b	86—biopsy-naive, prior negative	Mean 64.24 (±6.97)	Mean 10.00 (±8.53)	Mean 51.03 (±25.24)	1.5 3.0	≥3	PI-RADS unclassified—9 PI-RADS 3—22 PI-RADS 4—55 PI-RADS 5—30	GS > 3 + 3
Mischinger et al., 2018 [32]	Retrospective, 2b	130—biopsy-naive 72—prior negative	Median 66 (±7.6; IQR 60–73)	Median 8 (±5.8; IQR 6–11.9)	Median 36 (±21.8; IQR 26.9–47.8)	1.5 3,0	1–5	PI-RADS 1—1 PI-RADS 2—38 PI-RADS 3—39 PI-RADS 4—107 PI-RADS 5—17	GS ≥ 3 + 4
Kaufmann et al., 2018 [29]	Prospective, nonrandomized, 3b	156—prior negative	Median 67.0 (IQR 61.0–72.0)	Median 9.0 (IQR 6.0–13.0)	NA	3.0	NA	NA	GS ≥ 3 + 4 GS 3 + 3 MCCL ≥ 5 mm
Patel et al., 2020 [33]	Retrospective, 2b	92—biopsy-naive	Median 63 (IQR 58–68)	NA	Median 53.0 (IQR 41.5–75.5)	3.0	3–5	PI-RADS 3—28 PI-RADS 4—41 PI-RADS 5—22	ISUP ≥ 2
Yang et al., 2020 [34]	Prospective, nonrandomized, 2b	5—biopsy-naive 25—prior negative	Median 66 (range 53–80)	Median 8.1 (range 4.2–20.6)	Median 40.0 (range 18.6–70.0)	NA	3–5	NA	Epstein criteria
Lee et al., 2020 [30]	Retrospective, 2b	67—active surveillance 288—biopsy naive 145—prior negative	Mean 66.1 (±7.8)	Mean 10.4 (±8.3)	Mean 43.2 (±18.4)	3.0	3–5	PI-RADS 3—92 PI-RADS 4—288 PI-RADS 5—120	ISUP ≥ 2

GS—Gleason score, MCCL—maximum cancer core length.

**Table 2 cancers-15-01181-t002:** Primary and secondary outcomes of robot-assisted prostate biopsy.

Author, Year	Type of RA-TB	Software	Hardware	Approach	Number of Cores	Comparator, Number of Cores	csCDR	oCDR	CDR per Core	CDR per Lesion	Additional Utility	Procedure Duration, min	Complication
Kroenig et al., 2016 [16]	FUS-TB NA NA	Urofusion Urobiopsy	iSR’obot Mona Lisa (Biobot Surgical)	TP	Mean 10.2 (±4.8), median 9.0 (IQR 6.0–14.0)	RA-SB (Ginsburg study scheme), mean 30.0 (±5.6), median 32.0 (IQR 24.0–32.0)	23/52 (44.2%) RA-TB 26/52 (50.0%) RA-SB 27/52 (51.9%) Combined RA	26/52 (50.0%) RA-TB 31/52 (59.6%) RA-SB 31/52 (59.6%) Combined RA	NA	NA	2 (3.9%) PCa RA-SB	NA	1/52 (1.9%)—rectal perforation 1/52 (1.9%)— rectal bleeding
Kaufmann et al., 2016 [28]	FUS-TB First Unblinded	Urofusion Urobiopsy	iSR’obot Mona Lisa (Biobot Surgical)	TP	4	RA-SB, 14	28/55 (50.9%) RA-TB 14/55 (25.4%) RA-SB 29/55 (52.7%) Combined RA	29/55 (52.7%) RA-TB 19/55 (34.5%) RA-SB 34/55 (61.8%) Combined RA	90/220 (41%) RA-TB 37/770 (5%) RA-SB	NA	5 (9.1%) PCa RA-SB 1 (1.8%) csPCa RA-SB	Mean 43 (±6)	CD 1—9/55 (16.3%) 3/55 (5.4%)—bladder catheterization 1/55 (1.8%)—prolonged bleeding 5/55 (9.1%)—mild hematuria
Chen et al., 2017 [27]	COG-TB Second Blinded	Urofusion Urobiopsy	iSR’obot Mona Lisa (Biobot Surgical)	TP	Mean 13.3 (±5.8)	RA-SB, mean 26.9 (± 8.2)	4/18 (22.2%) RA-TB 3/18 (16.6%) RA-SB 5/18 (27.8%) Combined RA	10/18 (55.6%) RA-TB 14/18 (77.7%) RA-SB 14/18 (77.7%) Combined RA	28/239 (11.7%) RA-TB 33/511 (6.4%) RA-SB	NA	1 (5.5%) PCa RA-SB	Median 15 (range 13–21)	No severe complications
Miah et al., 2018 [31]	FUS-TB First NA	Urofusion Urobiopsy	iSR’obot Mona Lisa (Biobot Surgical)	TP	Mean 8.15 (±3.82)	RA-SB (Barzell scheme), mean 20.20 (±6.18)	35/86 (40.1%) RA-TB 44/86 (51.2%) Combined RA	NA	NA	48/116 (41.8%) Combined RA	9 (10.5%) csPCa RA-SB	NA	1/86 (1.1%)—urosepsis
Mischinger et al., 2018 [32]	FUS-TB First Unblinded	Urofusion Urobiopsy	iSR’obot Mona Lisa (Biobot Surgical)	TP	Mean 5.8 (±2.8)	RA-SB, mean 14.2	84/202 (41.6%) RA-TB 86/202 (42.6%) RA-SB 105/202 (52.0%) Combined RA	95/202 (47.0%) RA-TB 100/202 (49.5%) RA-SB 123/202 (60.9%) Combined RA	322/1192 (27.0%) RA-TB 316/2872 (11.0%) RA-SB	NA	28 (13.8%) PCa RA-SB 21 (10.4%) csCPa RA-SB	Mean 43 (±6)	CD ≤ II—6/202 (3.0%) 3/202 (1.5%)—AUR 3/202 (1.5%)—hematoma CD ≥ III —1/202 (0.5%) 1/202 (0.5%)—rectal injury, peritonitis
Kaufmann et al., 2018 [29]	FUS-TB NA NA	Urofusion Urobiopsy	iSR’obot Mona Lisa (Biobot Surgical)	TP	Median 3.0 (IQR 2.0–5.0)	MRI-TB manually, median 3.0 (IQR 2.0–5.0) TRUS COG-TB, median 3.0 (IQR 2.0–5.0)	26/73 (35.6%) RA-TB 18/45 (40%) %) In-bore manually 9/38 (23.6%) Cognitive TB	39/73 (53.4%) RA-TB 23/45 (51.1%) In-bore manually 11/38 (28.9%) Cognitive TB	NA	NA	NA	NA	No severe complications
Patel et al., 2020 [33]	FUS-TB First NA	Urofusion Urobiopsy	iSR’obot Mona Lisa (Biobot Surgical)	TP	Median 4 (IQR 3–5)	RA-SB (Ginsburg protocol), 24 Cognitive registration TB, median 3 (IQR 2–3) + SB (Ginsburg protocol), 24	17/53 (32.1%) RA-TB 21/53 (39.6%) Combined RA 4/39 (10.3%) Cognitive TB 14/39 (35.9%) Combined Cognitive TB + SB	25/53 (47.2%) RA-TB 32/53 (60.2%) Combined RA 5/39 (12.8%) Cognitive TB 17/39 (43.6%) Combined cognitive TB + SB	NA	NA	4 (7.5%) csPCa SB-RA	Median 24 (IQR 21–28) Combined RA Median 32 (IQR 31–36) Combined cognitive TB + SB	Combined RA 1/53 (1.9%)—AUR 4/53 (7.5%) hematuria Combined cognitive TB + SB 5/39 (12.8%)—AUR 15/39 (38.5%) hematuria
Yang et al., 2020 [34]	FUS-TB NA NA	Urofusion Urobiopsy	iSR’obot Mona Lisa (Biobot Surgical)	TP	Median 8 (range 5–16)	RA-SB, median 21 (range 9–48)	16/30 (53.3%) Combined RA	19/30 (63.3%) Combined RA	NA	NA	NA	Median 33 (range 19–49)	CD II—2/30 (6.7%) 1/30 (3.3%)—UTI 1/30 (3.3%)—AUR
Lee et al., 2020 [30]	FUS-TB First Unblinded	Urofusion Urobiopsy	iSR’obot Mona Lisa (Biobot Surgical)	TP	Mean 9.5 (±3.9)	RA-SB, mean 24.5 (±7.7)	166/433 (38.3%) RA-TB 171/433 (39.5%) RA-SB 199/433 (44.9%) Combined RA Active surveillance 20/67 (29.8%) RA-TB 21/67 (31.3%) RA-SB 23/67 (34.3%) Combined RA	213/433 (49.2%) RA-TB 222/433 (50.8%) RA-SB 248/433 (57.3%) Combined RA Active surveillance 40/67 (59.7%) RA-TB 41/67 (61.2%) RA-SB 48/67 (71.6%) Combined RA	NA	NA	NA	NA	NA

RA-TB—robot-assisted targeted biopsy; RA-TB—robot-assisted systematic biopsy; CD—Clavien–Dindo; NA—not available; AUR—acute urinary retention; UTI—urinary tract infection; MCCL—maximum cancer core length; CDR—cancer detection rate; csCDR—clinically significant cancer detection rate; PCa—prostate cancer; csPCa—clinically significant prostate cancer; MRI-TB—in-bore MR-targeted biopsy; FUS-TB–MRI-TRUS fusion targeted biopsy; COG-TB—cognitive registration TRUS-targeted biopsy.

## Data Availability

Data are available in MEDLINE/PubMed, Scopus.

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
