# Peer review of "Robot-Assisted Magnetic Resonance Imaging-Targeted versus Systematic Prostate Biopsy; Systematic Review and Meta-Analysis"

_cancers, 2023, doi:10.3390/cancers15041181_

Round 1

Reviewer 1 Report

Robot-assisted magnetic resonance imaging-targeted versus systematic prostate biopsy. Systematic review and meta-analysis.

The authors should be congratulated for a minacious review of the literature. These are some suggestions:

1.     I suggest using CSPCa as endpoint (primary outcome)

2.     Please add the template used for SB to the tables 

3.     Please consider which biopsy was performed first TB or SB, and if the operator performing SB was blinded to MRI results. 

4.     It is important to stratify by PIRADS and, if possible, by lesion location (anterior or posterior; lateral or medial; and base, mid or apex)

5.     Why did you exclude these data? “The data regarding patients on active surveillance was excluded.”  Would it be possible to identify and selectively exclude the data for men that were on AS? Or did you exclude all studies reporting men on AS?

6.     Should this study be excluded since it doesn’t have a RA-SB arm? “one study assessed transperineal RA- FUS-TB robot-assisted fusion biopsy compared to COG-TB, MRI-TB [29]”

7.     Studies including PIRADS 1-2 should be excluded, or at least, the data from PIRADS 1-2 patients should be excluded because TB isn’t performed on these patients.

8.     These are very large range on number of cores: “Median or mean number of cores for RA-TB ranged from 1.5 to 13.3 and from 14.0 to 32.0 for RA-SB.” These look like saturation TB (13.3 cores) and template mapping SB (32 cores). Please clarify on methods and tables.

9.     It would be ideal to include only GG>1 as definition of CSPCa

10.  Please keep the conclusions according to the methods and results. Your study doesn’t support these: “A robotic biopsy system can possibly reduce the influence of human factors on the results, that has high reproducibility with a high cancer detection rate and can potentially be performed under local anaesthesia.”

Thank you.

Reviewer 2 Report

The authors summarized the detection rate of robotic-assisted prostate biopsy and showed that this newer technique can be performed without major technical problems. As noted in the Limitation section, the number of biopsies performed in each study varied widely, and each was a small-size study. Therefore, it may be difficult to emphasize the difference and significance of this technique compared to previous techniques, especially fusion biopsy. Although well reviewed, I think it is needed to be careful about what is stated about differences from other techniques.

1. In the Discussion section paragraph 1, you point out that the robotic standardized approach in SB is one of the reasons for the lack of difference in the detection rate of csPCa, is that true? The detection rate of csPCa in RA-SB is not so high compared to previous reports.

2. In the Discussion section paragraph 1, the high number of TBs may not be the reason why cisPCa was detected in TBs as much as in SBs. In a per-patient or per-lesion analysis, if the quality of the MRI is assured and the number of biopsies is high, the detection rate of csPCa will increase and the detection rate of cisPCa will decrease. The accuracy of robotic targeting should also be suspected yet, according to these results.
